# Convergent changes in muscle metabolism depend on duration of high-altitude ancestry across Andean waterfowl

Neal J Dawson[1,2]*, Luis Alza[2,3,4], Gabriele Nandal[1], Graham R Scott[1], Kevin G McCracken[2,3,4,5,6]

[1]Department of Biology, McMaster University, Hamilton, Canada; [2]Department of Biology University of Miami, Coral Gables, United States; [3]University of Alaska Museum and Institute of Arctic Biology, University of Alaska Fairbanks, Fairbanks, United States; [4]Centro de Ornitología y Biodiversidad - CORBIDI, Lima, Peru; [5]Department of Marine Biology and Ecology, Rosenstiel School of Marine and Atmospheric Sciences, University of Miami, Miami, United States; [6]Human Genetics and Genomics, Hussman Institute for Human Genomics, University of Miami Miller School of Medicine, Miami, United States

**Abstract** High-altitude environments require that animals meet the metabolic $O_2$ demands for locomotion and thermogenesis in $O_2$-thin air, but the degree to which convergent metabolic changes have arisen across independent high-altitude lineages or the speed at which such changes arise is unclear. We examined seven high-altitude waterfowl that have inhabited the Andes (3812–4806 m elevation) over varying evolutionary time scales, to elucidate changes in biochemical pathways of energy metabolism in flight muscle relative to low-altitude sister taxa. Convergent changes across high-altitude taxa included increased hydroxyacyl-coa dehydrogenase and succinate dehydrogenase activities, decreased lactate dehydrogenase, pyruvate kinase, creatine kinase, and cytochrome c oxidase activities, and increased myoglobin content. ATP synthase activity increased in only the longest established high-altitude taxa, whereas hexokinase activity increased in only newly established taxa. Therefore, changes in pathways of lipid oxidation, glycolysis, and mitochondrial oxidative phosphorylation are common strategies to cope with high-altitude hypoxia, but some changes require longer evolutionary time to arise.

**\*For correspondence:**
neal.dawson@glasgow.ac.uk

**Competing interests:** The authors declare that no competing interests exist.

## Introduction

Given a common set of environmental challenges, evolution often converges upon a phenotype that maximizes fitness in that environment (i.e., fitness optimum). Many studies have explored the phenomenon of convergent evolution at molecular and biochemical levels by focusing on a single protein or gene across a broad number of taxa (*Storz, 2016*; *Storz et al., 2010*). However, we know relatively little about convergence of biochemical pathways, or how long convergent adaptations to an environment may take to evolve. Moreover, when evolution converges upon the same predictable phenotypes, does this process occur quickly over short evolutionary time scales or does it take longer durations to evolve?

Birds that have adapted to the challenges of high altitude present a compelling system in which to explore the convergence of metabolic pathways in response to common environmental challenges. The cold and hypoxic environment at high altitude requires that endotherms maintain high rates of $O_2$ consumption for locomotion and thermogenesis in $O_2$-thin air (*Bishop et al., 2015*;

*Hayes, 1989*). Flying birds face the additional challenge of maintaining lift with reductions in air density, which more than offsets the metabolic savings from reductions in drag, such that birds flying at high altitude must flap their wings harder and maintain higher metabolic rates to stay aloft (*Bishop et al., 2015*). Both evolved and phenotypically plastic changes in respiratory physiology and metabolism are believed to help mitigate the challenges posed by the cold and hypoxic environment at high altitude (*Beall, 2000*; *Lague et al., 2017*; *Monge and León-Velarde, 1991*). In the bar-headed goose (*Anser indicus*), for example, evolutionary adaptations to high altitude appear to have arisen throughout the $O_2$ transport pathway, including increases in effective ventilation, vital capacity and air-sac volume, haemoglobin-$O_2$ affinity, capillarity of the flight muscle and heart, and oxidative capacity of the flight muscle (*Jessen et al., 1991*; *McCracken et al., 2009a*; *Natarajan et al., 2015*; *Petschow et al., 1977*; *Scott et al., 2009a*; *Scott and Milsom, 2006*; *Scott and Milsom, 2007*; *Scott et al., 2011*; *Weibel, 1984*; *York et al., 2017*; *Zhang et al., 1996*). However, except for studies of a few key proteins like hemoglobin (*Natarajan et al., 2018*; *Natarajan et al., 2015*; *Projecto-Garcia et al., 2013*; *Storz et al., 2010*), we still know little about whether convergent phenotypic changes have arisen across independent high-altitude lineages, particularly for the pathways of energy metabolism that support locomotion and thermogenesis. Metabolic genes have been outliers in genome scans of selection in high-altitude taxa (*Qu et al., 2015*), and recent studies in high-altitude populations of mice and humans point towards skeletal muscle as a common target of selection (*Lundby and Calbet, 2016*; *Scott et al., 2018*). However, the extent to which convergent reorganization of metabolic pathways has occurred across high-altitude taxa to help sustain locomotion and thermogenesis in hypoxia remains unclear, particularly across species that independently colonized high altitude in the same geographic region.

The activities of enzymes involved in energy metabolism are important determinants of capacity and flux of metabolic pathways (*Kurata et al., 2007*; *Madhukar et al., 2015*; *Vogt et al., 2002a*; *Vogt et al., 2002b*). Flux capacity is an emergent property of the contributions of several enzymes in a pathway that is set via hierarchical regulation, which determines the limits of metabolic fluxes that can be achieved via metabolic regulation (*Suarez and Moyes, 2012*), and the maximal activities of key enzymes can be valuable markers of flux capacity. Some previous studies suggest that the activities of some enzymes in major energy producing pathways differ in high-altitude natives compared to their low-altitude counterparts (*Dawson et al., 2016*; *Leon-Velarde et al., 1993*; *Reyna-farje, 1962*; *Rosser and Hochachka, 1993*). However, similar differences in enzymatic activities have not been observed in other studies of high-altitude natives (*Mathieu-Costello, 2001*; *Scott et al., 2018*), and most previous work has been limited to a small number of enzymes and/or a single or a handful of species.

Here, we surveyed variation in 13 metabolic enzymes and myoglobin content across seven species encompassing four genera of high-altitude waterfowl (Family *Anatidae*) (*Figure 1*). Established markers of key metabolic pathways were chosen to provide a holistic view of energy metabolism during locomotion and thermogenesis, including aerobic and anaerobic glycolysis, fatty acid oxidation and mitochondrial function. We relied primarily on paired-lineage tests to make comparisons between high- and low-altitude taxa within a phylogenetic framework (*Storz et al., 2010*), but we also complemented these tests with standard ANOVA and phylogenetically independent contrasts (*Felsenstein, 1985*; *Garland et al., 2005*). We uncovered significant patterns of convergence in the remodeling of energy metabolism pathways in the major locomotor and thermogenic muscle, the pectoralis, between high- and low-altitude populations (summarized in *Figure 1*). Furthermore, by integrating population genetic data to infer how long each species has been established at high altitude, we show that some high-altitude phenotypes arose quickly whereas others required much longer evolutionary time to arise.

## Results and discussion

### Diversity in the duration of high-altitude ancestry

We collected muscle samples of birds from a broad range of high-altitude sites in the Andes and from paired low-altitude sites, from species in the genera *Anas* (n = 76 specimens), *Lophonetta* (n = 21 specimens), *Chloephaga* (n = 20 specimens) and *Oxyura* (n = 16 specimens). Our sampling effort included seven Andean waterfowl species, subspecies, or populations that independently

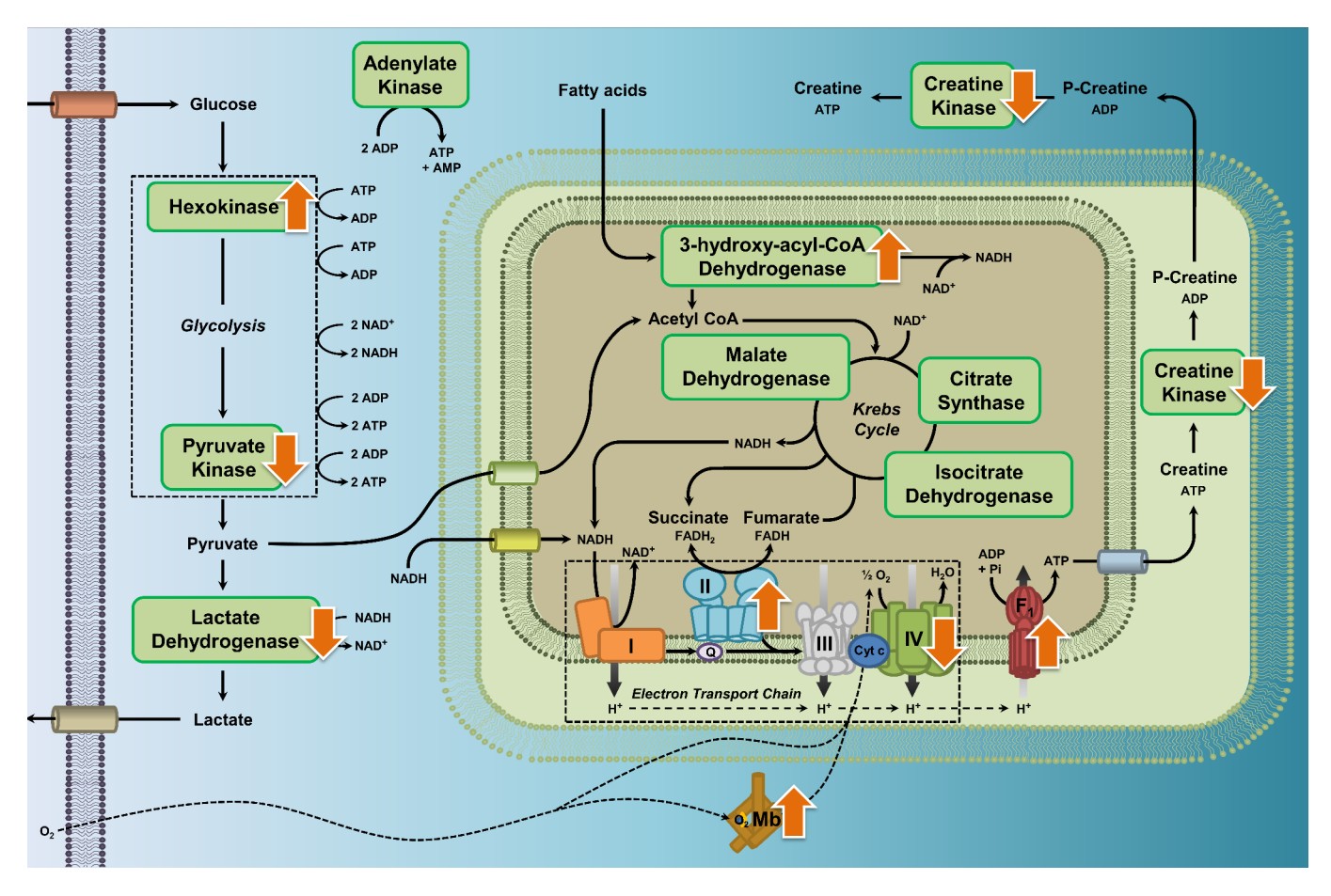

**Figure 1.** Enzyme pathway diagram illustrating where we observed differences in metabolic enzyme activity and myoglobin content in high-altitude waterfowl compared to their close low-altitude relatives. In addition to the observed increases in myoglobin content, increases in the activities of hexokinase, ATP synthase, HOAD, and complex II (succinate dehydrogenase), and decreases in activities of pyruvate kinase, lactate dehydrogenase, creatine kinase and complex IV (cytochrome c oxidase), we observed no changes in activity for the enzymes citrate synthase, isocitrate dehydrogenase, malate dehydrogenase, complex I (NADH-ubiquinone oxidoreductase), and adenylate kinase.

colonized high altitude at different times in geological history, and exhibit a range of divergence from their corresponding low-altitude population (*Figure 2*; *Table 1*). The species include strictly high-alpine specialists such as Andean goose (*C. melanoptera*, high-altitude range = 2000–5000 m above sea level) and puna teal (*Anas puna*, syn. *Spatula puna*, 3500–4600 m), which have diverged sufficiently from their low-altitude counterparts, Magellan goose (*C. picta*) and silver teal (*Anas versicolor*, syn. *Spatula versicolor*), to be classified as separate species (*Fjdelsa and Krabbe, 1990*). More recently diverged high-altitude populations of three dabbling duck species are classified as distinct subspecies, including crested duck (*L. specularioides alticola*, high-altitude range = 2000–5000 m; *L. s. specularioides*, low-altitude resident), speckled teal (*A. flavoristris oxyptera*, high-altitude range = 2500–4500 m; *A. f. flavirostris*, low-altitude resident), and cinnamon teal (*Anas cyanoptera orinomus*, syn. *Spatula cyanoptera orinomus*, high-altitude range = 2500–5000 m; *Anas cyanoptera*, syn. *Spatula cyanoptera*, low-altitude resident). The yellow-billed pintail (*A. georgia spinicauda*) has distinct populations occupying high altitude (up to 3500–4600 m) and low altitude, but they are not considered to be separate subspecies. Finally, we sampled a highly specialized diving duck that feeds extensively on aquatic insects, the ruddy duck, which has distinct subspecies at high altitude (*O. jamaicensis ferruginea*, high-altitude range = 2500–4500 m) and low altitude (*O. j. jamaicensis*).

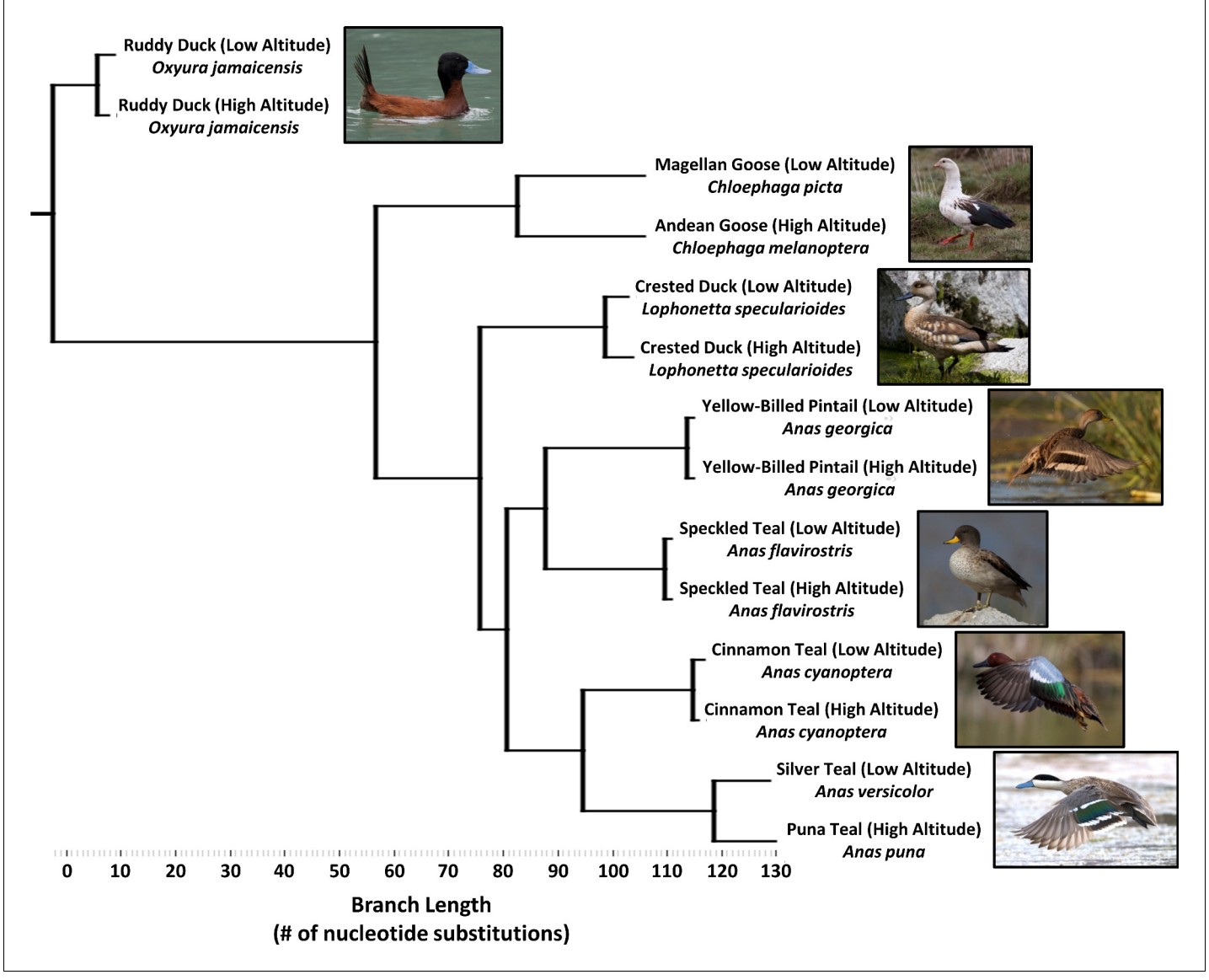

**Figure 2.** Simplified phylogenetic tree, generated using maximum parsimony and constrained to the same topology as the global waterfowl phylogeny published by *Gonzalez et al., 2009*. (see *Figure 2—figure supplement 1*). Branch lengths are measured as the total number of nucleotide substitutions in the 5' end of the mtDNA control region.

The online version of this article includes the following figure supplement(s) for figure 2:

**Figure supplement 1.** Phylogeny of the waterfowl based on Gonzalez et al.

In sum, two taxon pairs represent deeply diverged sister species living at high and low altitude, four taxon pairs (including the diving duck) represent intermediate divergence between subspecies within species, and the last pair represents shallow divergence between populations of the same subspecies (*Figure 2*, *Table 1*). To quantify duration of high-altitude ancestry, for each taxon pair we used previously published sequences from the mitochondrial DNA (mtDNA) control region to calculate population genetic parameters including: (a) the fixation index $\Phi_{ST}$, which measures nucleotide diversity ($\pi$) differences reflecting population subdivision, and (b) time since divergence between high and low altitude, as measured using a coalescent model incorporating drift and gene flow (*Hey, 2005*; *Hey and Nielsen, 2004*). We thus were able to reconstruct the rank order in time that these populations separated from each respective ancestral low-altitude population and order them according to the time they may have first become established in the Andean highlands.

**Table 1.** Seven species of Andean ducks showing classification level, $\Phi_{ST}$, time since divergence (t/site), and the approximate time (T) ago in years they became established at high altitude based on coalescent analysis.

$\Phi_{ST}$ and t/site were calculated using previously published mtDNA sequences. T in years was calculated using the substitution rate published by **Peters et al., 2005** of $4.8 \times 10^{-8}$ substitutions/site/year.

| Cinnamon teal | Yellow-billed pintail | Ruddy duck | Crested duck | Puna teal (H) Silver teal (L) | Speckled teal | Andean goose (H) Magellan goose (L) |
|---|---|---|---|---|---|---|
| New | New | New | Intermediate | Established | Established | Established |
| Subspecies | Populations | Subspecies | Subspecies | Species | Subspecies | Species |
| $\Phi_{ST} = 0.07$ | $\Phi_{ST} = 0.05$ | $\Phi_{ST} = 0.38$ | $\Phi_{ST} = 0.85$ | $\Phi_{ST} = 0.93$ | $\Phi_{ST} = 0.77$ | $\Phi_{ST} = 1.0$ |
| t/site = 0.000143116 | t/site = 0.00052227 | t/site = 0.000806087 | t/site = 0.003174242 | t/site = 0.017886364 | t/site = 0.019886364 | t/site = 0.04547956 |
| T (years) = 2982 | T (years) = 10,898 | T (years) = 16,793 | T (years) = 66,130 | T (years) = 372,633 | T (years) = 414,219 | T (years) = 947,491 |
| Capture range HA = 3812 m LA = 0–13 m | Capture range HA = 3812 m LA = 3 m | Capture range HA = 3812 m LA = 480–507 m | Capture range HA = 4281–4655 m LA = 760–1050 m | Capture range HA = 3812 m LA = 410–485 m | Capture range HA = 4209–4657 m LA = 760–1050 m | Capture range HA = 4368–4806 m LA = 0–27 m |
| HA (n = 8) LA (n = 8) | HA (n = 8) LA (n = 10) | HA (n = 6) LA (n = 10) | HA (n = 12) LA (n = 10) | HA (n = 11) LA (n = 10) | HA (n = 11) LA (n = 10) | HA (n = 12) LA (n = 8) |

## Convergence across multiple pathways of energy metabolism

There were convergent decreases in the activities of multiple glycolytic enzymes (LDH and PK) and in creatine kinase (CK) across high-altitude taxa (*Figure 3*). The reductions in CK activity in particular provided one of the strongest cases for convergent changes in high-altitude waterfowl (>50% reduction in highland populations in all species). High-altitude taxa exhibited a significant reduction in CK activity using Wilcoxon signed-rank test (p<0.001; *Figure 3*). There was also a significant main effect of altitude (p<0.0001; *Supplementary file 1b*) using two-factor ANOVA and a negative correlation between CK activity and altitude using phylogenetically independent contrasts (PICs) (p<0.0001; *Supplementary file 1f*). These changes could have been a plastic response to chronic hypoxia, as observed in humans exposed to high altitude (*Levett et al., 2015*; *Viganò et al., 2008*). Since most of the CK in muscle is cytosolic, reductions in CK activity likely reflect a reduction in cytosolic ATP buffering capacity and/or a shift towards a more oxidative phenotype. CK is also expressed in mitochondria of muscle, where it is specialized for high-energy phosphate transfer as part of the phosphocreatine shuttle, a process that could also be impaired if mitochondrial CK activity is reduced in high-altitude waterfowl. However, the activity of the phosphocreatine shuttle is augmented, not reduced, in high-altitude bar-headed geese, and mitochondrial CK expression is elevated in high-altitude deer mice (*Lui et al., 2015*; *Scott et al., 2009b*). These results suggest that decreases in CK activity, along with reductions in LDH and PK activities, may be part of a general strategy to downregulate some contributors to substrate-level phosphorylation in the muscle of high-altitude waterfowl. However, hexokinase (HK) activity was elevated in some of the least established high-altitude taxa (*Figure 3D*; *Supplementary file 1f*), so the decreases in LDH and PK activities are not associated with general reductions in capacity across glycolysis.

Several high-altitude waterfowl also exhibited increases in HOAD activity (1.2 to 2.2-fold), which likely increases the capacity for beta-oxidation of fatty acids (*Figure 3*). HOAD activity was significantly elevated in highland taxa using Wilcoxon's signed-rank test (p<0.001; *Figure 3*), and there was a significant main effect of altitude using two-factor ANOVA (p<0.0001; *Supplementary file 1b*) and a positive correlation between HOAD activity and altitude using PIC (p=0.0001; *Supplementary file 1g*). The peroxisome proliferator-activated receptors (PPAR) are key regulators of the expression of genes encoding the mitochondrial trifunctional enzyme (the heterooctamer that catalyzes the HOAD step and two additional steps in beta-oxidation) and other enzymes in beta-oxidation (i.e., acyl-CoA dehydrogenases), as well as genes controlling fatty-acid transport into mitochondria (*Fan and Evans, 2015*), so increases in HOAD activity in the flight muscle could reflect a general increase in the capacity for lipid oxidation. However, fatty-acid oxidation is strongly regulated by the enzymes involved in mitochondrial lipid uptake (e.g., carnitine palmitoyl transferase),

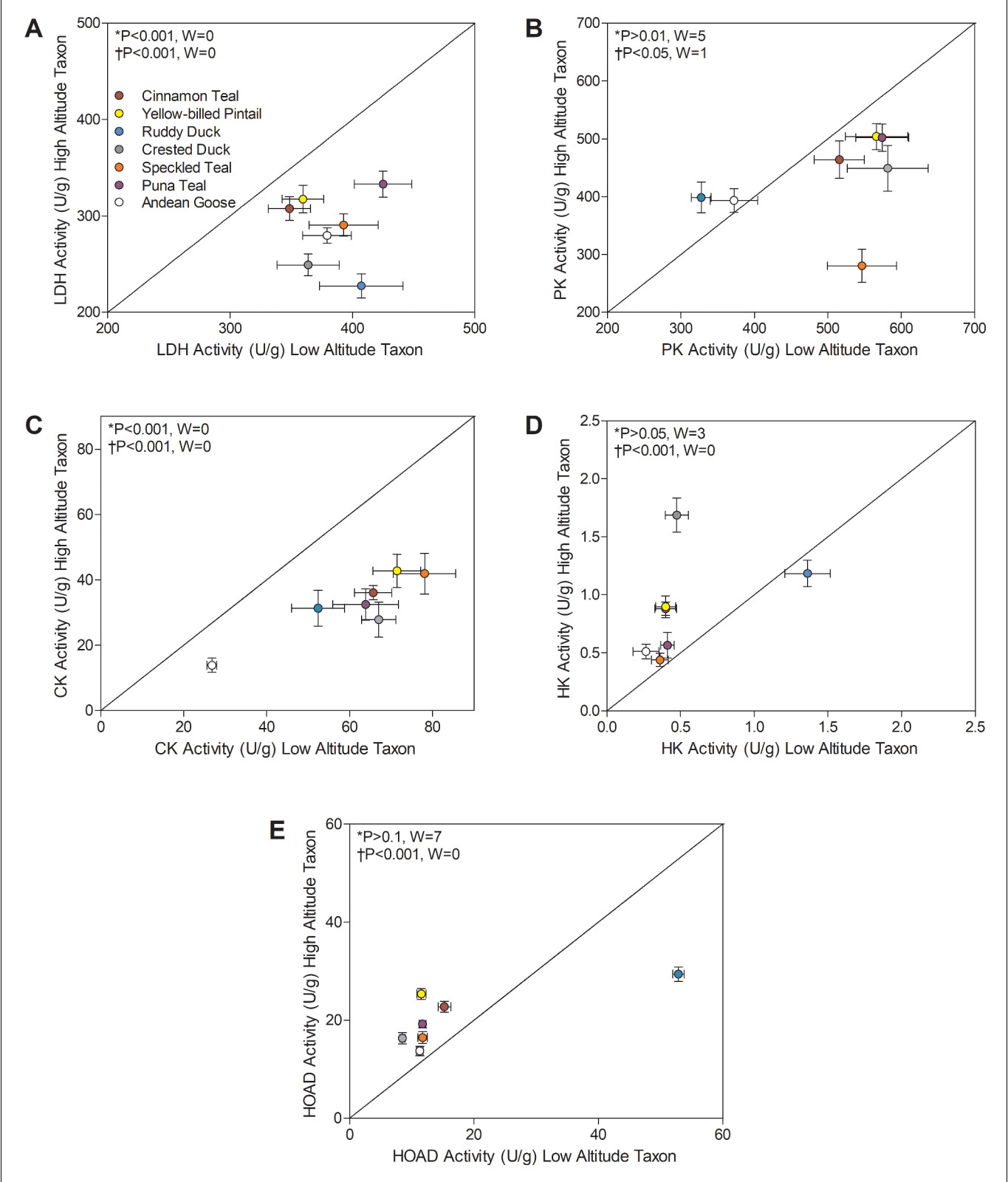

**Figure 3.** Metabolic enzyme activities for (**A**) lactate dehydrogenase (LDH), (**B**) pyruvate kinase (PK), (**C**) creatine kinase (CK), (**D**) hexokinase (HK), and (**E**) 3-hydroxyacyl-CoA dehydrogenase (HOAD), measured in the pectoralis of high- and low-altitude waterfowl. The diagonal represents the line of equality (x = y). Values are shown as mean ± SEM U/g tissue (*n* = 8–12). High-altitude values are significantly different overall from the corresponding low-altitude values when p<0.05 in Wilcoxon's Signed-Rank Tests, which were carried out including (*) and excluding (†) ruddy ducks.

such that changes in fatty acid oxidation can arise without changes in HOAD activity (*Morash et al., 2013*). Nevertheless, if increases in HOAD activity are indeed associated with increased capacity for lipid oxidation in the flight muscle, such changes would amplify the already remarkable capacity of birds to support high rates of muscle metabolism and power on lipids alone (*Guglielmo, 2010*; *O'Brien and Suarez, 2001*; *Suarez et al., 1986*). Sustained thermogenesis relies heavily on lipid oxidation (*Marsh and Dawson, 1989*; *Swanson and Thomas, 2007*; *Vaillancourt et al., 2009*; *Vaillancourt et al., 2005*), such that cold temperatures could have increased lipid metabolism and stimulated a corresponding rise in beta-oxidation capacity in high-altitude birds. Furthermore, there seems to be a positive association between altitude and body lipid content in insects (*Parkash et al., 2008*; *Sømme et al., 1996*), which could increase dietary lipid availability at high altitude for waterfowl (most of which forage on aquatic insects). There is also evidence showing a positive correlation between dietary lipids and HOAD activity in the pectoralis of both migratory and non-migratory birds (*Guglielmo, 2010*; *Maillet and Weber, 2007*; *Nagahuedi et al., 2009*). If high-altitude birds eat more lipid-rich foods than their low-altitude counterparts, then a corresponding increase in lipid metabolism might have stimulated the rise in beta-oxidation capacity.

There were also convergent changes in complexes II and IV of the electron transport system and in ATP synthase ($F_1F_O$-ATPase, complex V) (*Figure 4*). The activities of all TCA cycle enzymes assayed, including citrate synthase, isocitrate dehydrogenase, and malate dehydrogenase, were similar between high- and low-altitude populations (*Supplementary file 1a*). Citrate synthase in particular is a common marker of mitochondrial volume density in muscle tissue (*Boushel et al., 2007*; *Dawson et al., 2018*; *Larsen et al., 2016*; *Mahalingam et al., 2017*; *Mogensen et al., 2006*), so this result suggests that muscle mitochondrial content was similar between high- and low-altitude populations. The observed changes in complexes II, IV, and V activities could therefore reflect a change in mitochondrial quality affecting the function of a given amount of mitochondria. Complex II activity was higher in multiple high-altitude species (1.05- to 1.51-fold increases, with no change in complex I activity; *Supplementary file 1a*) using Wilcoxon's signed-rank test ($p<0.05$; *Figure 3*), and there was a significant main effect of altitude using two-factor ANOVA ($p=0.0060$; *Supplementary file 1b*) and a positive correlation between complex II activity and altitude using PIC ($p=0.0213$; *Supplementary file 1f*). ATP synthase activity was also higher in multiple (but not all) high-altitude taxa using Wilcoxon's signed-rank test ($p<0.05$; *Figure 4*), supported by results of two-factor ANOVA (altitude effect, $p<0.0001$; *Supplementary file 1b*) and the positive correlation between ATP synthase activity and altitude using PIC ($p=0.0009$; *Supplementary file 1f*). In contrast, the terminal acceptor for oxygen, cytochrome c oxidase (COX; complex IV), had ~50% lower activity when compared to low-altitude sister taxa across nearly all high-altitude species except the ruddy duck. In fact, there seemed to be a narrow optimum for the activity of COX across high-altitude waterfowl, as all species converged on a strikingly similar value (*Figure 4B*). COX activity showed a significant reduction in highland taxa using Wilcoxon's signed-rank test ($p<0.05$; *Figure 3*), and there was a significant main effect of altitude in two-factor ANOVA ($p<0.0001$; *Supplementary file 1b*) and a negative correlation between COX activity and altitude using PIC ($p=0.0001$; *Supplementary file 1f*). Unique specializations in the activity, structure and function of COX have been observed in the locomotory muscles of several high-altitude taxa (*Dawson et al., 2016*; *Lui et al., 2015*; *Scott et al., 2011*; *Sheafor, 2003*). A similar reduction in COX activity (~50% less) was also observed in the cardiac muscle of bar-headed goose compared to low-altitude geese, in association with an increased affinity for cytochrome c (*Scott et al., 2011*). Similarly, COX of some hypoxia-tolerant fish has decreased activity but a greater affinity for $O_2$ (*Lau et al., 2017*). Therefore, hypoxia may drive convergent changes in COX and mitochondrial function across vertebrates.

Several high-altitude waterfowl exhibited elevated myoglobin in the pectoralis muscle when compared to low-altitude ducks. Myoglobin content showed a significant main effect of altitude ($p=0.0077$; *Supplementary file 1b*) using two-factor ANOVA and this effect was nearly significant in Wilcoxon's signed-rank test ($0.1 > P > 0.05$; *Figure 5B*) and in PIC correlations between myoglobin content and altitude ($p=0.0706$; *Supplementary file 1g*). Myoglobin content was variable across species and tended to be greatest in ducks with the highest body mass (*Supplementary file 1a*), and the effect of body mass was nearly significant ($p=0.0585$; *Supplementary file 1d*). Body mass, however, was not significantly greater in high-altitude waterfowl ($p=0.9707$; *Supplementary file 1b*), suggesting that the elevated levels of myoglobin in some high-altitude taxa is not simply due to increased body size. Elevated myoglobin content in flight muscle may serve to increase cellular $O_2$

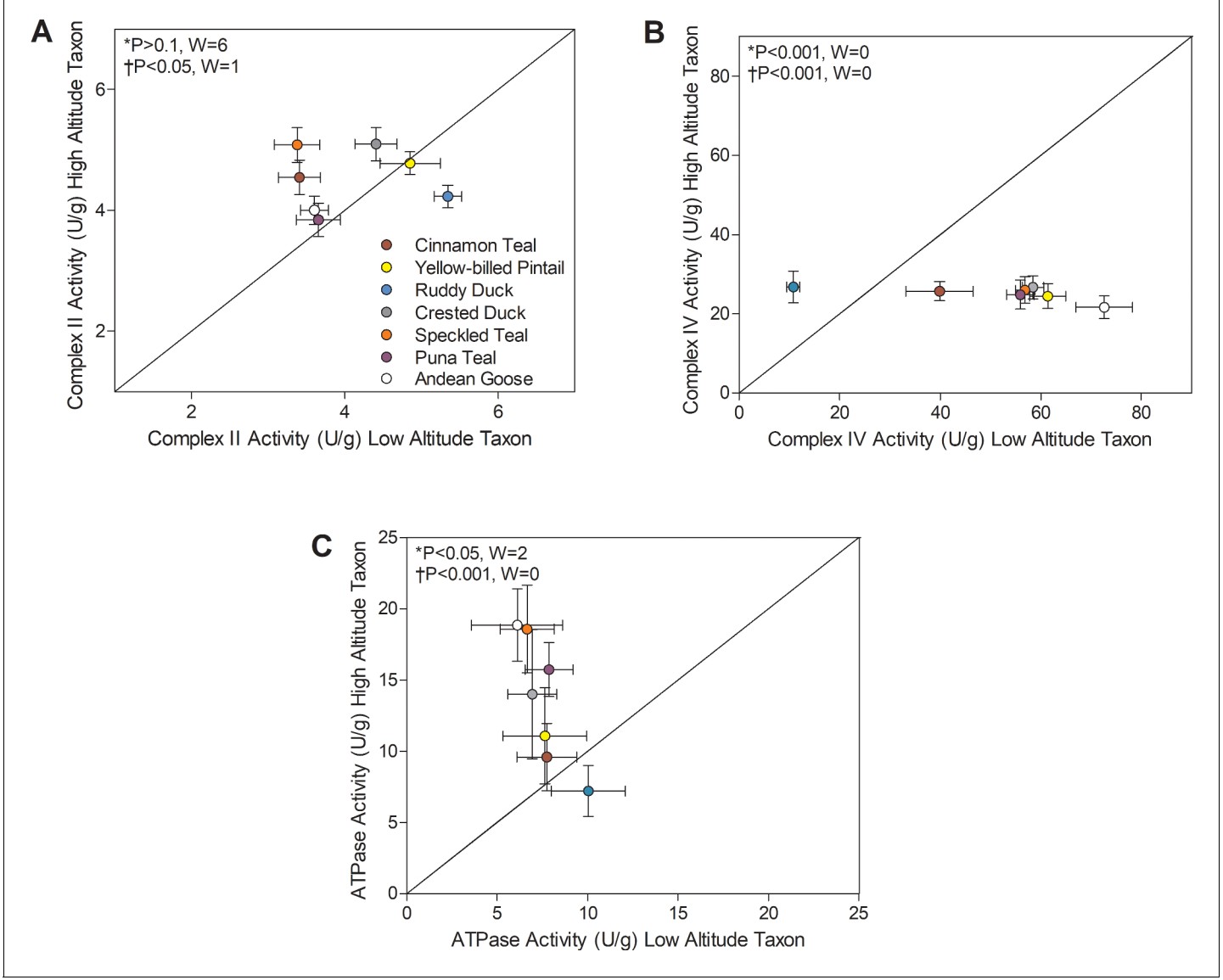

**Figure 4.** Mitochondrial enzyme activities for (**A**) Complex II, (**B**) Complex IV, and (**C**) ATP synthase measured in the pectoralis of high- and low-altitude waterfowl. The diagonal represents the line of equality (x = y). Values are shown as mean ± SEM U/g tissue (*n* = 8–12). High-altitude values are significantly different overall from the corresponding low-altitude values when p<0.05 in Wilcoxon's Signed-Rank Tests, which were carried out including (*) and excluding (†) ruddy ducks.

stores and to facilitate intracellular $O_2$ diffusion (*Kanatous et al., 2009*; *Wittenberg and Wittenberg, 2003*). Elevated myoglobin content could potentially augment intracellular lipid transport as well, as there are some suggestions that myoglobin may bind and facilitate fatty acid diffusion through the sarcoplasm (*Gros et al., 2010*). Elevated myoglobin content or transcript expression has been previously observed in some other taxa that reside at and/or were acclimatized to high altitude, including torrent ducks (*Mergantta armata*) (*Dawson et al., 2016*), dogs (*Gimenez et al., 1977*), rats (*Vaughan and Pace, 1956*), and Tibetan humans (*Moore et al., 2002*). Therefore, elevated myoglobin levels in the locomotory muscle appear to be an important strategy across high-altitude taxa for augmenting mitochondrial $O_2$ availability (and possibly intracellular lipid transport) and thus sustaining thermogenesis and locomotion in hypoxia at high altitude.

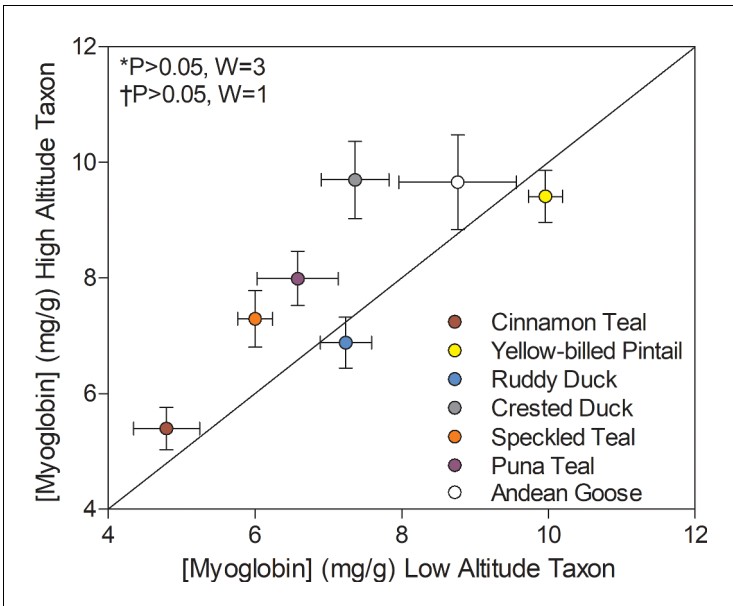

**Figure 5.** Myoglobin content measured in the pectoralis of high- and low-altitude waterfowl. The diagonal represents the line of equality (x = y). Values are shown as mean ± SEM mg/g tissue (*n* = 8–12). High-altitude values are significantly different overall from the corresponding low-altitude values when p<0.05 in Wilcoxon's Signed-Rank Tests, which were carried out including (*) and excluding (†) ruddy ducks.

## Idiosyncratic changes in high-altitude ruddy ducks

In many cases, the ruddy duck, the most distantly related and the only diving species studied herein, showed a contrasting pattern of changes compared to the other species that forage on the surface of water (dabbling ducks) or graze on land (sheldgeese). Low-altitude ruddy ducks seem to have fundamentally different physiological and biochemical characteristics than the other low-altitude taxa in this study, with relatively high activities of HOAD, complex II, and HK, relatively low activities of complex IV (*Figures 3–6*), and a relatively high haemoglobin-$O_2$ affinity atypical of low-altitude populations but more similar to high-altitude waterfowl populations (*Natarajan et al., 2015*). Compared to this low-altitude relative, high-altitude ruddy ducks had decreased activities of both complex II and HOAD in the flight muscle, in contrast to most/all other high-altitude taxa in which the activities of these enzymes were elevated (*Figure 3C*; *Figure 4A*), and they also do not show the typical and expected increase in Hb-$O_2$ affinity that is seen in the other high-altitude waterfowl from this study system (*Natarajan et al., 2015*) and in many other high-altitude birds (*Storz, 2016*). The distinct direction of the differences between high- and low-altitude ruddy ducks appeared to strongly contribute to the significant species × altitude interactions that were detected in two-factor ANOVAs for PK (p=0.0003), HOAD (p<0.0001), complex II (p<0.0001), and complex IV (p<0. 001) activities (*Supplementary file 1b*). It is possible that diving created a unique set of physiological challenges for this species as it invaded high-altitude habitats; however, we do not see similar changes in enzyme activities in the pectoralis (flight muscle) or gastrocnemius (swimming muscle) of torrent ducks at high altitude, another diving duck species native to the Andes (*Dawson et al., 2016*). It is more likely that the unique features of low-altitude ruddy ducks – related to diet, physiology, biochemistry, etc. – favoured distinct mechanisms of adaptation or plasticity during the process of high-altitude colonization.

## Some metabolic changes arise only after prolonged evolutionary time at high altitude

One of the clear advantages to these particular high-altitude waterfowl as a study system is the ability to infer and rank order the evolutionary time these populations have lived at high altitude, and thus provide insight into metabolic changes that take longer evolutionary times to arise. Using population genetic data for the mtDNA control region, we determined that the high-altitude taxa studied

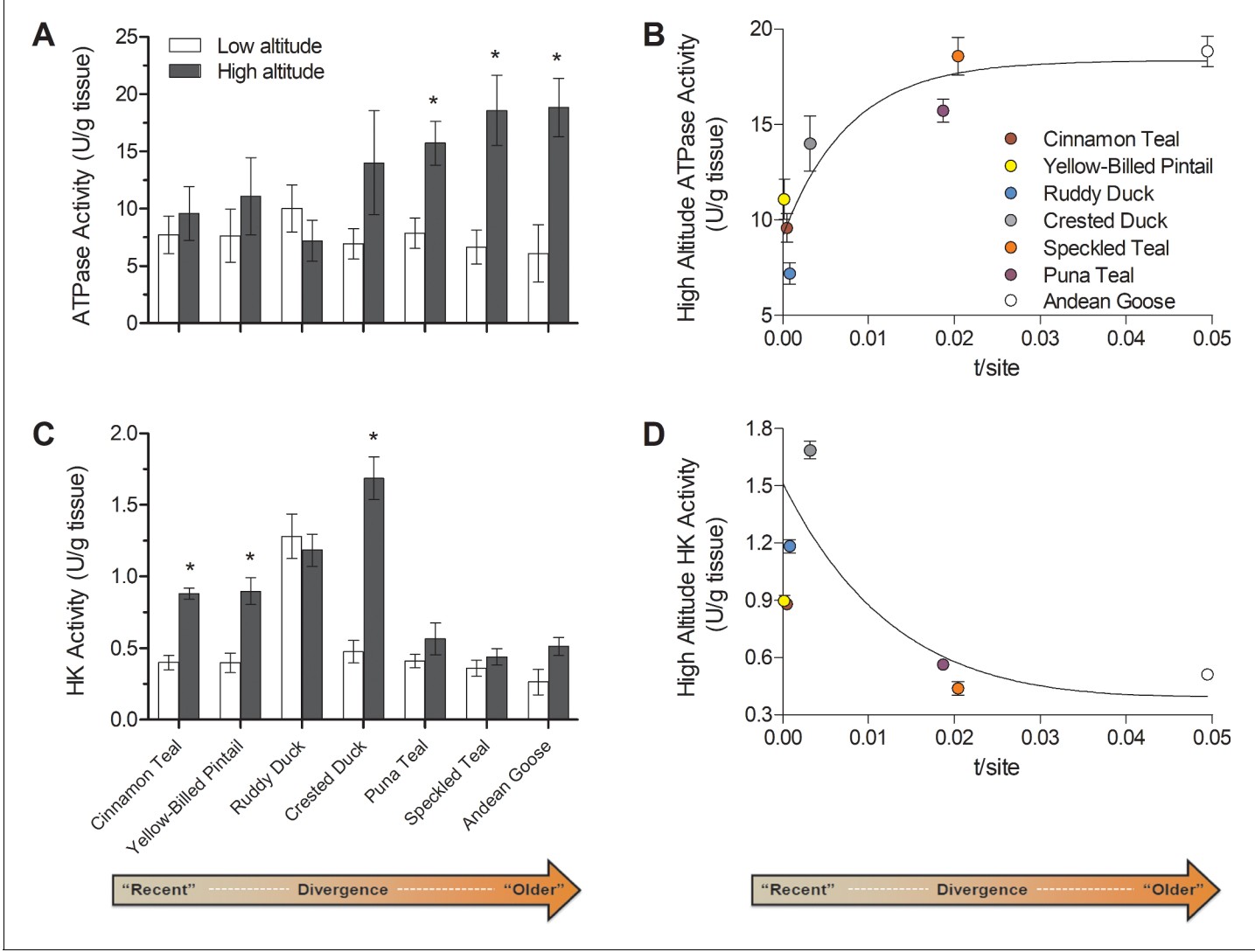

**Figure 6.** Changes over evolutionary time at altitude of (**A**) ATP synthase activity and (**C**) hexokinase activity measured in the pectoralis of seven high- and low-altitude waterfowl pairs. Values are given as the mean ± SEM U/g tissue (n = 8–12). * - Significantly different activity in high-altitude ducks compared to low-altitude ducks (two-factor ANOVA followed by the Bonferroni post-tests; p<0.05). (**B**) ATP synthase and (**D**) Hexokinase activities in each high-altitude taxon plotted against the t/site value between each high-low pair.

here most likely diverged from their low-altitude sister taxa for evolutionary times that differed by as much as three orders of magnitude, from within the last several thousands of years (i.e., more recently established high-altitude populations) to approximately a million years in the case of species that are now established high-altitude endemic species (*Table 1*). In each case, all available evidence indicates that high-altitude taxa were derived from low-altitude ancestral populations and not the reverse (*Bulgarella et al., 2014*; *Bulgarella et al., 2012*; *Fjeldså, 1985*; *Graham et al., 2018*; *McCracken et al., 2009b*; *McCracken et al., 2009c*; *Muñoz-Fuentes et al., 2013*; *Wilson et al., 2013*). This approach allowed us to broadly classify high-altitude taxa into three groups based on their mtDNA divergence: newly established highland populations (~3,000–17,000 years; cinnamon teal, yellow-billed pintail and ruddy duck), intermediate (66,000 years; crested duck), and long established subspecies/species (372,000–947,000 years; puna teal, speckled teal, Andean goose) (*Table 1*). These time since divergence (T) calculations, while approximations, are derived from t/site (τ) values representing the average tree height of all genealogies in each coalescent analysis divided by the mutation rate (μ), assuming a mutation rate of $4.8 \times 10^{-8}$ substitutions per site per year

(*Peters et al., 2005*) as previously applied to these same species. In support of our inferences about duration of high-altitude ancestry, a very similar rank order is also seen with $\Phi_{ST}$ (*Table 1*), which provides an independent estimate of divergence not dependent on any particular mutation rate. Using this rank order, we predicted that some metabolic pathways would only change in the more established lineages (i.e., deeper T, higher $\Phi_{ST}$) that have occupied the high-altitude environment for longer and have therefore experienced hypoxia for longer evolutionary time, therefore with more time for adaptation to proceed.

Following this classification, ATP synthase activity was elevated in only the most established high-altitude taxa, whereas no differences were observed in the newly established populations (*Figure 6A,B*). The magnitude of these increases in established high-altitude taxa were appreciable, ranging from 2.0- to 3.1-fold, and they appeared to strongly contribute to the significant species × altitude interaction that was detected in two-factor ANOVA (p=0.0342; *Supplementary file 1b*). These changes would have led to a strong increase in the capacity for ATP synthesis relative to electron transport. Such a change could reduce mitochondrial membrane potential and attenuate the production of reactive oxygen species (ROS), which may be advantageous at high altitude for reducing oxidative stress (*Brand et al., 1999*; *Korshunov et al., 1997*; *Skulachev, 1996*). Increasing ATP synthesis capacity may also reduce the magnitude of phosphorylation control over mitochondrial respiration, shifting more control towards the electron transport system (*Gnaiger et al., 2000*; *Jacobs et al., 2012*; *Pesta and Gnaiger, 2012*).

Our results also show that newly established populations at high altitude show increased HK activity in comparison to low-altitude populations (*Figure 3D*; *Figure 6C,D*). Most of the low-altitude taxa had very similar HK activities in the flight muscle (with the exception of ruddy ducks). Only newly established and intermediate high-altitude populations had elevated HK activities, ranging from 2.20- to 3.56-fold, which likely contributed to the significant species × altitude interaction that was detected in two-factor ANOVA (p<0.0001; *Supplementary file 1b*). These transient increases in HK activity along with the convergent reductions in PK and LDH activity suggest that there are broad changes across glycolysis in high-altitude taxa, potentially associated with changes in carbohydrate oxidation and/or lactate production. Hypoxia exposure can lead to plastic increases in the reliance on carbohydrate oxidation in mammals (*Hochachka et al., 1998*; *McClelland et al., 1998*; *Robin et al., 1984*) and may be useful at high altitude by generating more ATP per molecule of $O_2$ than lipid oxidation. However, carbohydrate oxidation may be constrained over time by glycogen stores if carbohydrate fuels cannot be adequately supplied by the circulation (*McClelland, 2004*), and we observed that the rise in HK activity in newly-established populations returned to low-altitude levels in the most established high-altitude taxa (*Figure 6C,D*). These particular high-altitude taxa may instead take advantage of more plentiful but $O_2$-costly fuels (i.e., lipids), provided that sufficient tissue $O_2$ supply is maintained by evolved or plastic changes throughout the $O_2$ transport pathway (e.g., increases in haemoglobin $O_2$ affinity, expansion of pulmonary $O_2$ diffusing capacity) (*Maina et al., 2017*; *Natarajan et al., 2015*).

## Conclusions

Convergent changes have occurred in many pathways of metabolism across high-altitude waterfowl, with increases in capacity for beta oxidation of lipids and adjustments in the activity of oxidative phosphorylation (OXPHOS) enzymes that likely fine-tune mitochondrial function (*Figure 1*). However, some changes required longer evolutionary time at high altitude to arise, suggesting that adaptive changes in high-altitude taxa may involve several steps, such that some changes are only observed in the longest-established highland taxa. Indeed, some changes in the activities of enzymes involved in metabolizing lipids and carbohydrates may be convergent across taxa (increased HOAD activity, decreased PK and LDH activities), whereas others are more time-dependent. In the latter case, HK activity is elevated in relatively new high-altitude colonists, but is subsequently blunted over evolutionary time. Similar distinctions between convergent (e.g., reduced complex IV activity) and time-dependent (increase in complex V activity in only the most established highland taxa) changes exist for mitochondrial OXPHOS enzymes. It is likely that high-altitude animals rely upon hypoxia acclimatization (*Hochachka et al., 1998*) when first colonising high altitude, followed by evolved physiological specializations that adjust the capacity and flux of metabolic pathways, along with evolved improvements in mitochondrial $O_2$ supply that arise from increases in tissue capillarity (*Leon-Velarde et al., 1993*; *Mathieu-Costello et al., 1998*; *Scott et al., 2009a*), Hb-$O_2$ affinity

(*Galen et al., 2015*; *Natarajan et al., 2015*; *Projecto-Garcia et al., 2013*), circulatory O$_2$ delivery, and/or pulmonary O$_2$ uptake (*Calbet et al., 2003*; *Maina et al., 2017*; *McClelland and Scott, 2019*). Our data suggest that increases in capacity for beta oxidation, changes in capacity across glycolysis, and adjustments in mitochondrial function are common strategies to cope with the challenges of high altitudes, but that longer time scales of evolutionary adaptation can be required to fully converge upon the ultimate high-altitude phenotype.

# Materials and methods

## Key resources table

| Reagent type (species) or resource | Designation | Source or reference | Identifiers | Additional information |
|---|---|---|---|---|
| Chemical compound, drug | Glucose | Sigma G8270 | D-(+)-Glucose ≥99.5% (GC) | Enzyme assay reagent |
| Chemical compound, drug | ATP | Sigma A2383 | Adenosine 5'-triphosphate disodium salt hydrate Grade I,≥99%, from microbial | Enzyme assay reagent |
| Chemical compound, drug | MgCl$_2$ | Sigma M8266 | Magnesium Chloride anhydrous,≥98% | Enzyme assay reagent |
| Chemical compound, drug | NADP$^+$ | BioShop Canada NAD007 | B-NADP, Disodium trihydrate,>95% | Enzyme assay reagent |
| Chemical compound, drug | G6PDH | Roche 10127655001 | Glucose-6-Phosphate Dehydrogenase (G6P-DH) grade I, from yeast | Enzyme assay reagent |
| Chemical compound, drug | LDH | Roche 10127876001 | L-Lactate Dehydrogenase (L-LDH) from rabbit muscle | Enzyme assay reagent |
| Chemical compound, drug | PEP | Sigma P7002 | Phosphoenolpyruvic acid trisodium salt hydrate ≥97% (enzymatic) | Enzyme assay reagent |
| Chemical compound, drug | ADP | Sigma A5285 | Adenosine 5'-diphosphate monopotassium salt dehydrate bacterial, ≥95%, powder | Enzyme assay reagent |
| Chemical compound, drug | Pyruvate | Sigma P2256 | Sodium pyruvate ReagentPlus,≥99% | Enzyme assay reagent |
| Chemical compound, drug | NADH | BioShop Canada NAD002 | NADH ß-NICOTINAMIDE ADENINE REDUCED | Enzyme assay reagent |
| Chemical compound, drug | Oxaloacetate | Sigma O4126 | Oxaloacetic acid ≥97% (HPLC) | Enzyme assay reagent |
| Chemical compound, drug | Acetyl CoA | BioShop Canada ACO201 | ACETYL COENZYME A, Trilithium Salt | Enzyme assay reagent |
| Chemical compound, drug | DTNB | Sigma D218200 | 5,5'-Dithiobis (2-nitrobenzoic acid) ReagentPlus, 99% | Enzyme assay reagent |
| Chemical compound, drug | Isocitrate | Sigma I1252 | DL-Isocitric acid trisodium salt hydrate ≥93% | Enzyme assay reagent |
| Chemical compound, drug | CoQ$_{10}$ | Sigma C9538 | Coenzyme Q10, ≥98% (HPLC) | Enzyme assay reagent |
| Chemical compound, drug | Rotenone | Sigma R8875 | Rotenone, ≥95% | Enzyme assay reagent |
| Chemical compound, drug | BSA | Sigma A6003 | Bovine Serum Albumin lyophilized powder, essentially fatty acid free,≥96% (agarose gel electrophoresis) | Enzyme assay reagent |

*Continued on next page*

*Continued*

| Reagent type (species) or resource | Designation | Source or reference | Identifiers | Additional information |
|---|---|---|---|---|
| Chemical compound, drug | KCN | Sigma 60178 | Potassium cyanide BioUltra,$\geq$98.0% (AT) | Enzyme assay reagent |
| Chemical compound, drug | Succinate | Sigma S2378 | Sodium succinate dibasic hexahydrate ReagentPlus,$\geq$99% | Enzyme assay reagent |
| Chemical compound, drug | DCPIP | Sigma D1878 | 2,6-Dichloroindophenol sodium salt hydrate, BioReagent | Enzyme assay reagent |
| Chemical compound, drug | DUB | Sigma D7911 | Decylubiquinone, $\geq$97% (HPLC) | Enzyme assay reagent |
| Chemical compound, drug | CytcCH$_2$ | Sigma C7752 | Cytochrome c from equine heart $\geq$95% based on Mol. Wt. 12,384 basis | Enzyme assay reagent |
| Chemical compound, drug | Oligomycin | Sigma O4876 | Oligomycin from *Streptomyces diastatochromogenes* $\geq$90% total oligomycins basis (HPLC) | Enzyme assay reagent |
| Chemical compound, drug | HK | Roche 11426362001 | Hexokinase (HK) | Enzyme assay reagent |
| Chemical compound, drug | Acetoacetyl CoA | Sigma A1625 | Acetoacetyl coenzyme A sodium salt hydrate Cofactor, for acyl transfer | Enzyme assay reagent |
| Chemical compound, drug | Creatine | Sigma C3630 | Creatine monohydrate, $\geq$98% | Enzyme assay reagent |
| Chemical compound, drug | PK | Roche PK-RO | Pyruvate Kinase (PK) from rabbit muscle | Enzyme assay reagent |
| Chemical compound, drug | KH$_2$PO$_4$ | P5378 | Potassium phosphate monobasic, ReagentPlus | Assay buffer reagent |
| Chemical compound, drug | EGTA | Sigma E4378 | Ethylene glycol-bis (2-aminoethylether)-N,N,N',N'-tetraacetic acid, $\geq$97.0% | Assay buffer reagent |
| Chemical compound, drug | EDTA | Sigma EDS | Ethylenediamin etetraacetic acid BioUltra, anhydrous, $\geq$99% (titration) | Assay buffer reagent |
| Chemical compound, drug | Triton-X 100 | Sigma X100 | Triton X-100 laboratory grade | Assay buffer reagent |
| Software, algorithm | Geneious | Biometters Ltd., Auckland, NZ | | Used for sequence alignment |
| Software, algorithm | PAUP | Version 4, Sinauer Associates, Sunderland, Massachusetts, USA | | Used to generate branch lengths |
| Software, algorithm | MESQUITE | https://www.mesquiteproject.org/ | | Used to analyze phylogenetic contrasts |
| Software, algorithm | PDAP module | http://mesquiteproject.org/pdap_mesquite/ | | Used to analyze phylogenetic contrasts |
| Software, algorithm | IM | https://bio.cst.temple.edu/~hey/software | | Used to calculate divergence |

## Tissue sampling

Waterfowl were captured at high altitudes (3822–4806 m) or at low altitudes (0–1050 m) in various locations in and near the Andes across South America and from low-altitude sites in North America. Tissues from some birds were sampled immediately on site, whereas others were sampled after birds

were transported to a nearby field lab where they were provided with unlimited access to water for 12–18 hr, prior to being euthanized. In all cases, samples of pectoralis muscles were quickly dissected and frozen in liquid $N_2$ and stored at −80°C for enzyme analysis (see below). Muscle samples were taken at three depths in the middle of the tissue (surface, intermediate, and deep), in order to account for heterogeneity of muscle fibers throughout the pectoralis This heterogeneity is important to consider, because flight muscle tends to become more oxidative at deeper depths from the ventral surface (*Scott et al., 2009a*), as confirmed by the variation observed here (*Supplementary file 1j*).

Samples were imported to Canada with authorization from the Canadian Wildlife Service (Scientific Possession No. 369) and collected with authorization from the Servicio Nacional de Area Naturales Protegidas del Peru (004–2014-SERNANP-DGANP-RNT/J), Dirección General Forestal y de Fauna Silvestre del Peru (RD 169–2014 MIN AGRI-DGFFS/DGEFFS, 190–2015-SERFOR-DGGSPFFS), Ministerio de Industria, Agricultura, y Ganaderia Chubut (No. 24/07 y 1636/14), Ministerio de Asuntos Agrarios Buenos Aires, Ministerio de Producción de Entre Rios, Oregon Department of Fish and Wildlife (101-15), and USFWS Region 1 Migratory Bird Permit Office (MB68890B-0). All protocols were carried out in accordance with guidelines that were approved by the Institutional Animal Care and Use Committee at the University of Miami or University of Alaska.

## Phylogenetic tree generation

Mitochondrial DNA (mtDNA) sequences were obtained for each population in this study. Most were available from previously published data sets including *McCracken et al., 2009c* for *A. georgica*, *Bulgarella et al., 2012* for *Lophonetta specularioides*, *Wilson et al., 2013* for *A. cyanoptera* (syn. *Spatula cyanoptera*), *Muñoz-Fuentes et al., 2013* for *Oxyura jamaicensis*, *Bulgarella et al., 2014* for *Chloephaga* spp., and *Graham et al., 2018* for *A. flavirostris*. To this, we also supplemented unpublished mtDNA sequences from *A. puna* and *A. versicolor* (syn. *Spatula* spp.) (*Wilson et al., 2013*. Dissertation 2010). The sequence we utilized comprise ~684 bp of the mtDNA control region corresponding to previously published primers L78-H774 (*Johnson and Sorenson, 1999*; *Sorenson and Fleischer, 1996*). GenBank accession number can be found in the referenced articles and in supplementary materials (*Supplementary file 1i*). As each species possessed numerous small indels, alignment was performed in Geneious (Biomatters Ltd., Auckland, NZ). Indels were treated as missing data, and the resulting alignment was refined by eye to correct ambiguities. All sequences were obtained using PCR and capillary DNA sequencing protocols as described in *McCracken et al., 2009c*. Next, we generated a tree with branch lengths constrained to match the *Gonzalez et al., 2009* topology, which is the most recent phylogenetic analysis of waterfowl to include all of these species. Branch lengths for this tree were generated using maximum parsimony in the software PAUP (Version 4, Sinauer Associates, Sunderland, Massachusetts, USA).

## Enzyme activities and myoglobin assays

The maximal activities of 13 enzymes as well as myoglobin content were assayed as previously described (*Dawson et al., 2016*). The reported enzyme activity content for each individual bird is the average value from the samples taken across the three depths of muscle (which are fully reported in *Supplementary file 1j*). Wilcoxon signed-rank test (paired lineage tests; *Figures 3–5*) as well as two-factor ANOVA followed by Bonferroni post-tests (*Figure 6*; *Supplementary file 1b*, *Supplementary file 1c*) were used to make statistical comparisons between high-altitude versus low-altitude taxa. The maximal activities of 13 enzymes as well as myoglobin content were assayed as previously described (*Dawson et al., 2016*) using a Spectramax Plus 384 spectrophotomer (Molecular Devices, Sunnyvale, CA, USA) at avian body temperature of 41°C. Samples were homogenized in 10 volumes of ice-cold homogenizing buffer [100 mM $KH_2PO_4$ buffer, pH 7.2, containing 1 mM EGTA, 1 mM EDTA, 0.1% Triton-X 100, and 1 mM phenylmethylsulfonyl fluoride (PMSF)]. Homogenates were then centrifuged at 1,000 × g at 4°C and the supernatant was collected for use in assays. Enzyme activities were determined for each sample as the difference between the rate measured using all assay components (assayed in triplicate) and the background reaction rate, all measured at avian body temperature (41°C). Measurements were carried out as described in *Supplementary file 1h*. Preliminary experiments determined that all substrate concentrations were saturating. Enzyme activities are expressed in units of micromole substrate per gram tissue per

minute, with protein concentrations determined using the Bradford method (BioRad, Mississauga, ON, Canada). Myoglobin content was assayed in triplicate using a modified version of the original method (*Reynafarje, 1962*), as described in *Dawson et al., 2016*. Biochemicals were from Sigma-Aldrich (Oakville, ON, Canada) unless otherwise stated.

Wilcoxon signed-rank test (paired lineage tests; *Figures 3–5*) as well as two-factor ANOVA followed by Bonferroni post-tests (*Figure 6*; *Supplementary file 1b*, *Supplementary file 1c*) were used to make statistical comparisons between high-altitude versus low-altitude taxa. Data are presented as means ± SE. $p < 0.05$ was considered significant. When significant interactions occurred, they were often attributed primarily to opposing patterns observed in the ruddy ducks. The exceptions were HK and ATPase, for which the significant interactions were attributed to changes that appeared to be associated with differences in evolutionary time at high altitude.

The linear relationship between mass and enzyme activity or myoglobin content was assessed in order to determine if enzyme activity or myoglobin content varies with the mass of individual waterfowl independent of altitude. Enzyme activity or myoglobin content was plotted on the y-axis against mass on the x-axis from both high- and low-altitude waterfowl and the slope, as well as, goodness of fit ($r^2$) were determined (*Supplementary file 1d*, *Supplementary file 1e*). A significant relationship between mass and enzyme activity was determined when $p < 0.05$.

## Phylogenetically independent contrasts

To conduct analysis of phylogenetically independent correlations (PIC), the maximum parsimony tree constrained to match the same topology as *Gonzalez et al., 2009* global waterfowl phylogeny (Fig. S1) and branch lengths were imported into MESQUITE (*Maddison and Maddison, 2016*). The PDAP module (*Midford, 2010*) was used to carry out PIC analyses to assess whether the relationship between altitude and enzyme activity persisted after taking into account the effects of phylogeny (*Supplementary file 1f*, *Supplementary file 1g*). Significant relationships between raw contrasts and their standard deviations were not generally observed, and only in a few cases when ruddy ducks were included in the analysis. In such cases, we used exponential transformation of branch lengths to eliminate these significant relationships before carrying out PIC correlations (*Garland et al., 1992*). We observed similar results in PIC analyses using branch lengths that were set to one (data not shown).

## Time at altitude and time since divergence

To quantify duration of high-altitude ancestry, we used two methods for each taxon pair to calculate population genetic parameters indicative of the evolutionary time each highland population has likely been at high altitude, and to examine the time dependence of activity of enzymes. First, we calculated the pairwise fixation index $\Phi_{ST}$ between each pair of low- and high-altitude sister populations. We utilized the same previously published mtDNA control region data sets for these populations (*Bulgarella et al., 2014*; *Bulgarella et al., 2012*; *Graham et al., 2018*; *McCracken et al., 2009c*; *Muñoz-Fuentes et al., 2013*; *Wilson et al., 2013*). This index of population differentiation, comparable to Wright's (1965) fixation index $F_{ST}$, is bounded by 0 and 1 (*Wright, 1965*). Thus, whereas pairs of populations with $\Phi_{ST}$ closer to 0 are expected to be recently diverged, $\Phi_{ST}$ closer to 1 is indicative of populations with much older divergence, as is the case for fully differentiated species that have ceased gene flow. Populations with intermediate $\Phi_{ST}$ are expected to fall somewhere in the middle of this continuum. This approach allowed us to relate metabolic distinctiveness between high- and low-altitude populations to their relative magnitude of genetic divergence. Second, we used a two-population coalescent model in the software IM (Hey Lab, Temple University) to calculate the population divergence parameter, t/site, which represents the average tree height of all genealogies in each coalescent analysis (*Hey, 2005*; *Hey and Nielsen, 2004*). Multiplied by the mutation rate ($\mu$) this parameter can then be used to calculate time since divergence (T) in years. This has advantages over $\Phi_{ST}$ because it allowed us to incorporate evolutionary processes in our model including divergence due to genetic drift, and therefore uncertainty in our estimates, as well as gene flow (i.e., migration in both directions) for which some of these populations have been shown to exhibit more connectivity than others. Starting parameters for these analyses were conditioned on the data by first making several preliminary runs using wide priors and described as published previously (*Bulgarella et al., 2014*; *Bulgarella et al., 2012*; *Graham et al., 2018*;

*McCracken et al., 2009c*; *Muñoz-Fuentes et al., 2013*; *Wilson et al., 2013*). Finally, we were able to convert t/site to time in years using the substitution rate for the 5' end of the control region published by *Peters et al., 2005* of $4.8 \times 10^{-8}$ substitutions/site/year (*Peters et al., 2005*). This provided an approximation to the date at which these high-altitude populations might have experienced a founder event following the colonization of high altitude for the first time, whereas the rank order of $\Phi_{ST}$ also likely corresponds to their rank order newest to oldest high-altitude residents. Finally, it should be noted that the Andes have been rising throughout the Cenozoic for approximately 80 million years (*Ramos, 2005*; *Royden et al., 2008*), and while some parts of the Andes uplifted more recently than others, the oldest of these waterfowl populations (i.e., South American sheldgeese) are probably diverged no more than several millions years ago (*Johnson and Sorenson, 1999*). Therefore, in all cases the Andes had already uplifted to close to their present height when these high-altitude populations first became established.

## Acknowledgements

This research was supported by funds from McMaster University, the Canadian Foundation for Innovation, the Ontario Ministry of Research and Innovation, and a Natural Sciences and Engineering Research Council of Canada (NSERC) Discovery Grant to GRS. This research was also supported by a NSF grant (IOS-0949439) to KGM and the Kushlan Endowment for Waterbird Biology and Conservation at the University of Miami. Gratitude for all kinds of assistance is expressed to E Bautista, M Bulgarella, C, Kopuchian, D Lijtmaer, B Reishus, M Smith, M St. Louis, P Tubaro, and T Valqui. GRS is supported by the Canada Research Chairs Program, NJD was supported by an NSERC Postdoctoral Fellowship, and KGM is supported by the Kushlan Chair for Waterbird Biology and Conservation at the University of Miami.

## Additional information

### Funding

| Funder | Grant reference number | Author |
|---|---|---|
| Natural Sciences and Engineering Research Council of Canada | Discovery Grant | Graham R Scott |
| National Science Foundation | IOS-0949439 | Kevin G McCracken |
| Canada Foundation for Innovation | John R. Evans Leaders Fund | Graham R Scott |
| Ontario Ministry of Research and Innovation | Early Researcher Award | Graham R Scott |
| Kushlan Endowment for Waterbird Biology and Conservation | Kushlan Chair | Kevin G McCracken |
| Canada Research Chairs | Tier 2 - Comparative and Environmental Physiology | Graham R Scott |
| Natural Sciences and Engineering Research Council of Canada | Postdoctoral Fellowship | Neal J Dawson |

The funders had no role in study design, data collection and interpretation, or the decision to submit the work for publication.

### Author contributions

Neal J Dawson, Conceptualization, Data curation, Formal analysis, Validation, Investigation, Visualization, Methodology, Writing - original draft, Project administration, Writing - review and editing; Luis Alza, Resources, Project administration, Writing - review and editing; Gabriele Nandal, Investigation, Writing - review and editing; Graham R Scott, Kevin G McCracken, Conceptualization,

Resources, Supervision, Funding acquisition, Methodology, Project administration, Writing - review and editing

### Author ORCIDs
Neal J Dawson (ID) https://orcid.org/0000-0001-5389-8692
Graham R Scott (ID) http://orcid.org/0000-0002-4225-7475

### Ethics
Animal experimentation: Samples were imported to Canada with authorization from the Canadian Wildlife Service (Scientific Possession No. 369) and collected with authorization from the Servicio Nacional de Area Naturales Protegidas del Peru (004-2014-SERNANP-DGANP-RNT/J), Dirección General Forestal y de Fauna Silvestre del Peru (RD 169-2014-MIN AGRI-DGFFS/DGEFFS, 190-2015-SERFOR-DGGSPFFS), Ministerio de Industria, Agricultura, y Ganaderia Chubut (No. 24/07 y 1636/14), Ministerio de Asuntos Agrarios Buenos Aires, Oregon Department of Fish and Wildlife (101-15), and USFWS Region 1 Migratory Bird Permit Office (MB68890B-0). All protocols were carried out in accordance with guidelines that were approved by the institutional animal care and use committee at the University of Miami or University of Alaska.

### Decision letter and Author response
Decision letter https://doi.org/10.7554/eLife.56259.sa1
Author response https://doi.org/10.7554/eLife.56259.sa2

## Additional files

### Supplementary files
• Supplementary file 1. Supporting data tables, statistical analyses, and methodology. (**a**) Maximal activities (µmol/g tissue/min), body mass (g) and myoglobin (Mb; mg/g tissue) concentration in pectoralis muscle. (**b**) Two-factor ANOVA results of maximal activities (µmol/g tissue/min), mass (g) and myoglobin (Mb; mg/g tissue) concentration in pectoralis muscle. (**c**) Two-factor ANOVA results of maximal activities (µmol/g tissue/min), mass (g) and myoglobin (Mb; mg/g tissue) concentration in pectoralis muscle excluding data for ruddy ducks from the subfamily *Oxyurinae*. (**d**) Test of covariance for enzyme activity (µmol/g tissue/min) or myoglobin content (Mb; mg/g tissue) and body mass (g). (**e**) Test of covariance for enzyme activity (µmol/g tissue/min) or myoglobin content (Mb; mg/g tissue) and body mass (g) excluding data for ruddy ducks from the subfamily *Oxyurinae*. (**f**) Correlation analyses of phylogenetic independent contrasts of bird mass (g), myoglobin (Mb) content (mg/g tissue), or enzyme activity (µmol/g tissue/min) *versus* altitude (m). (**g**) Correlation analyses of phylogenetic independent contrasts of bird mass (g), myoglobin (Mb) content (mg/g tissue), or enzyme activity (µmol/g tissue/min) *versus* altitude (m) excluding data for ruddy ducks from the subfamily *Oxyurinae*. (**h**) Assay conditions for enzymatic measurements. (**i**) List of GenBank gene accession numbers for mtDNA control region used in the construction of the phylogenetic tree. (**j**) Maximal activities (µmol/g tissue/min) in pectoralis muscle from surface, intermediate and deep tissue sampling locations.

• Supplementary file 2. Complete list of accession numbers, databases and source references used to create the phylogenetic tree.

• Transparent reporting form

### Data availability
- Mitochondrial DNA sequences for Anas puna and Anas versicolor were deposited in GenBank under accession numbers MN734269-MN734345 (details in Supplementary file 2). All data generated or analysed during this study are included in the manuscript and supporting files.

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
