## [Decision Letter]

Thank you for submitting your article "Convergent changes in muscle physiology depend on duration of high-altitude ancestry across Andean waterfowl" for consideration by *eLife*. Your article has been reviewed by three peer reviewers, including Kevin Campbell as Guest Editor and Reviewer #1, and the evaluation has been overseen by Christian Rutz as the Senior Editor. The following individual involved in the review of your submission has agreed to reveal their identity: Andrew Murray (Reviewer #2).

The reviewers have discussed their reviews with one another, and the Guest Editor has drafted this decision letter to help you prepare a revised submission. We would like to draw your attention to changes in our revision policy that we have made in response to COVID-19 (https://elifesciences.org/articles/57162).

Summary:

This study employed 7 high- and low-altitude species pairs that spanned ~1 million years of evolution in hypoxic environments to identify convergent and time-dependent trajectories in muscle metabolic biochemistry to high altitude environments. Importantly, since the work examines fuel and energy metabolism in flight muscle, rather than oxygen transport, it provides a valuable perspective on what is happening at the level of metabolic biochemistry. Overall, the experiments appear to have been carefully conducted, analyses seem sound (except the construction of the phylogenetic tree, which requires particular attention during revision), and the manuscript is well written.

Essential revisions:

1) The precision of writing related to evolution, convergence, and phenotypic plasticity could be improved. The first paragraph doesn't fully account for the wide variety of evolutionary studies of convergence and other processes, and sets up a fairly weak hypothesis about whether convergence is always the same or idiosyncratic because it sets up a black and white dichotomy. Evolution/adaptation/drift are always happening and of course each lineage will have an idiosyncratic process. They may arrive at generally similar phenotypes, but do you ever expect exact matching? The second question about the speed of adaptation is more interesting. Accordingly, the first paragraph or most of it could be dropped and the paper could be started with paragraph 2.

2) The interpretation of the results pertaining to glycolytic capacity is handled somewhat inconsistently throughout the manuscript. The Abstract states that glycolytic capacity is decreased at altitude, whereas the data for HK activity in particular does not support this conclusion with Figure 1 indicating greater HK activity at high altitude. The argument made in the main text is more subtle, supporting an increase in HK activity in many of the more recent high-altitude populations, but no difference in the more established residents. In fact, HK activity is not lower in any highland waterfowl studied compared with its low altitude counterpart with the exception of the ruddy duck (Supplementary file 1A). Moreover, the difference between the low- and high-altitude Andean goose species is almost two-fold, though this appears not to be significantly different? Could the authors confirm this, as the data in Supplementary file 1A seem to suggest that it might be. The Abstract should be modified to better reflect the manuscript in regards to the above discussion, and would also benefit from the inclusion of additional details regarding what parameters were actually measured.

3) More information needs to be provided regarding the phylogenetic tree construction methodology, which is central for the phylogenetically independent contrast analysis outlined in the Materials and methods. For instance, what specific algorithm and settings were used in Geneious for the multiple sequence alignment? Was the resulting alignment refined by eye to correct ambiguities, and how were indels dealt with? Additionally, it is stated that the tree was constructed using "neighbor joining" in the Materials and methods (again, with key details/parameters missing), though it is noted that a "UPGMA tree and branch lengths" were used for the PIC analysis. Importantly, the tree presented in Figure 2 is poorly resolved, which may have affected the results of the PIC analyses (several of which were borderline significant). Since the results of the PIC analysis are only meaningful if the input tree is accurate/reliable, the authors should consider using more current and robust tree building methods (e.g., maximum likelihood) with the best-fitting models to better resolve the actual relationships among these species. Additionally, given the relatively low amount of sequence data employed for tree building, the authors should also consider including additional sequence data where available and constraining the positions of certain lineages based on the results of other studies with larger datasets. For example, constraining Loponetta as sister to Anas (see e.g. Bulgarella et al., 2010), and placing Oxyura as the outgroup (see e.g. Gonzalez et al., 2009), though we recommend using more recent/robust phylogenetic hypotheses if available.

Bulgarella, M., Sorenson, M.D., Peters, J.L., Wilson, R.E. and McCracken, K.G., 2010. Phylogenetic relationships of Amazonetta, Speculanas, Lophonetta, and Tachyeres: four morphologically divergent duck genera endemic to South America. Journal of Avian Biology, 41(2), pp.186-199.

4) The authors helpfully sampled from three sites in the muscle, at different depths, and took a mean of the activities from each location. Please report the degree of variation between the three locations.

5) It would be helpful for all data to be presented in the manner of Figures 3, 4, and 5. Figure 6 is useful in showing differences in HK activity and ATPase activity in relation to evolutionary time, but the data in panel 6D, which only shows absolute rates of HK activity for high altitude groups, suggest that HK activity decreases below low altitude values as a function of time, whereas it is higher than in lowland counterparts (albeit not always significantly higher) for almost all species studied.

6) One of the main themes of the paper was that fatty acid oxidation capacity is increased at high altitude. However, this is based on a single enzyme indicator, HOAD, whereas other aspects of metabolism were assessed with multiple enzymes. This potential limitation should be noted or discussed.

7) In the third paragraph of the subsection “Convergence Across Multiple Pathways of Energy Metabolism”, the change in complex II (succinate dehydrogenase) activity is attributed to a need to support greater capacities for fatty acid oxidation, though this is not correct. The free FADH2 from beta-oxidation does not donate electrons to complex II (which oxidizes succinate using a fixed FAD/FADH cofactor), but to the electron-transferring flavoprotein complex (CETF). Please clarify/revise.

8) Why are the results for Complex V (ATP synthase) not reported in the comparisons between populations/species, and included only in the analysis of time? They seem to be important.

9) The possible explanations for increased ATP-synthase activities at altitude are useful, but the second paragraph of the subsection “Some Metabolic Changes Arise Only After Prolonged Evolutionary Time at High Altitude” refers to anoxia, and does not appear to be relevant to non-anoxia tolerant waterfowl. The authors cite a review article here (Garcia-Aguilar and Cuevez) but the original work was carried out by St-Pierre and colleagues (2000, PNAS, 97(15), pp.8670-8674) in anoxia-tolerant frogs. The relevance to the current study is not clear, since it is not necessarily the case that membrane potential would fall in hypoxia exposed waterfowl (a different state to anoxia).

10) The authors speculate that the differences in metabolic function are related to hypoxia or cold at altitude, but given this is a flight muscle, how might the decreased air resistance alter the metabolic demand placed on this muscle? Would activity be expected to fall as there is less drag, or might the low air pressure increase the metabolic demand of lift?

11) In the statistical analyses presented in Supplementary file 1B and C, there are highly significant species by altitude interactions, but these are not addressed. When there are significant interactions, main effects should not be considered until the interactions are addressed, either by plotting them out to see if they are all in the same direction and just have different slopes, or by separating the species and looking at them individually if there is crossover of the effects. The authors need to discuss what the interactions were and how they handled them. Most of these are present in cases where a significant altitude effect was detected, so they are potentially crucial to the conclusions.

---

## [Author Response]

Summary:This study employed 7 high- and low-altitude species pairs that spanned ~1 million years of evolution in hypoxic environments to identify convergent and time-dependent trajectories in muscle metabolic biochemistry to high altitude environments. Importantly, since the work examines fuel and energy metabolism in flight muscle, rather than oxygen transport, it provides a valuable perspective on what is happening at the level of metabolic biochemistry. Overall, the experiments appear to have been carefully conducted, analyses seem sound (except the construction of the phylogenetic tree, which requires particular attention during revision), and the manuscript is well written.

We would like start by thanking the reviewers for their excellent suggestions. We feel as though the suggestions put forward by the reviewers have greatly improved the manuscript. We agree that the phylogenetic tree needed to be improved. We have reconstructed the phylogenetic tree by using maximum parsimony while constraining it to the topology depicted in Gonzalez, 2019. This has resolved the two polytomies present in our original tree and has resulted in minor changes in the reported statistics; however, there are no changes in the significance reported previously, largely due to the fact that each high-low pair was quite distantly related from all other pairs. Nevertheless, we believe the paired-lineage Wilcoxon tests are the most appropriate for examining the convergent differences that exist across high-low pairs. We have made some minor editorial changes throughout the Results to bring the results of the Wilcoxon tests to the forefront.

Essential revisions:1) The precision of writing related to evolution, convergence, and phenotypic plasticity could be improved. The first paragraph doesn't fully account for the wide variety of evolutionary studies of convergence and other processes, and sets up a fairly weak hypothesis about whether convergence is always the same or idiosyncratic because it sets up a black and white dichotomy. Evolution/adaptation/drift are always happening and of course each lineage will have an idiosyncratic process. They may arrive at generally similar phenotypes, but do you ever expect exact matching? The second question about the speed of adaptation is more interesting. Accordingly, the first paragraph or most of it could be dropped and the paper could be started with paragraph 2.

We agree that the question surrounding the speed at which adaptation may take place is more interesting, and as such, we have pared down the initial paragraph to focus on this particular question and have removed the first question regarding similar or idiosyncratic changes.

2) The interpretation of the results pertaining to glycolytic capacity is handled somewhat inconsistently throughout the manuscript. The Abstract states that glycolytic capacity is decreased at altitude, whereas the data for HK activity in particular does not support this conclusion with Figure 1 indicating greater HK activity at high altitude. The argument made in the main text is more subtle, supporting an increase in HK activity in many of the more recent high-altitude populations, but no difference in the more established residents. In fact, HK activity is not lower in any highland waterfowl studied compared with its low altitude counterpart with the exception of the ruddy duck (Supplementary file 1A). Moreover, the difference between the low- and high-altitude Andean goose species is almost two-fold, though this appears not to be significantly different? Could the authors confirm this, as the data in Supplementary file 1A seem to suggest that it might be. The Abstract should be modified to better reflect the manuscript in regards to the above discussion, and would also benefit from the inclusion of additional details regarding what parameters were actually measured.

We agree with the reviewers that the Abstract would benefit from being more precise and from including greater detail of the exact parameters measured. We have made the appropriate changes to the Abstract to reflect these suggestions. We realize in hindsight that we were inconsistent in how we discussed the issue of glycolytic capacity, both in the Abstract and elsewhere, and we have made several other changes to correct this inconsistency (Abstract, subsections “Convergence Across Multiple Pathways of Energy Metabolism”, “Some Metabolic Changes Arise Only After Prolonged Evolutionary Time at High Altitude”, and “Conclusions”).

To clarify, differences in HK activity between high- and low-altitude geese are not statistically significant. Even when doing a simple pairwise comparison using a student’s t-test, there is no significant difference between high- and low-altitude geese.

3) More information needs to be provided regarding the phylogenetic tree construction methodology, which is central for the phylogenetically independent contrast analysis outlined in the Materials and methods. For instance, what specific algorithm and settings were used in Geneious for the multiple sequence alignment?

The default settings used included treatment of gaps as missing data, which is appropriate for a gene such as the mtDNA control region in which it becomes difficult to code gaps of a different length as character states. The search criterion used for the simplified tree for the PICs was maximum parsimony, as the UPGMA method cannot accommodate a constraint. The maximum parsimony tree with branch lengths (Figure 2) was constrained to the Gonzalez (2009) topology, which is the most comprehensive global phylogeny yet available for *all* of these waterfowl species (Gonzalez, Düttman and Wink, 2009). To clarify their taxonomic position among the many lineages of waterfowl we have added a supplementary figure (Figure 2—figure supplement 1) showing their position in the Gonzalez et al., 2009 topology. This new figure clearly emphasizes the relatively distant phylogenetic relationships between each high-low pair, although there is a much closer relationship within each high-low pair. We have made a small revision in the Introduction to remove a statement that was inconsistent with this information.

Was the resulting alignment refined by eye to correct ambiguities, and how were indels dealt with?

The reviewers are correct that the resulting alignment was refined by eye to correct any ambiguities. We treated any indels as missing data (see above). We have included this information in the Materials and methods subsection “Phylogenetic Tree Generation”.

Additionally, it is stated that the tree was constructed using "neighbor joining" in the Materials and methods (again, with key details/parameters missing), though it is noted that a "UPGMA tree and branch lengths" were used for the PIC analysis.

We would like to thank the reviewer for pointing out this mistake. As stated above, we have reconstructed our phylogenetic tree using maximum parsimony constrained to the topology depicted in Gonzalez et al., 2009, which we now describe accurately in both the main text and Figure 2—figure supplement 1 (subsection “Phylogenetic Tree Generation”).

Importantly, the tree presented in Figure 2 is poorly resolved, which may have affected the results of the PIC analyses (several of which were borderline significant).

Our newly built tree using maximum parsimony has fully resolved the two polytomies present in our original tree constrained to match Gonzalez et al., 2009. This has only resulted in minor changes in the reported statistics, however, such that there are no qualitative differences and no changes in the significance reported previously.

Since the results of the PIC analysis are only meaningful if the input tree is accurate/reliable, the authors should consider using more current and robust tree building methods (e.g., maximum likelihood) with the best-fitting models to better resolve the actual relationships among these species. Additionally, given the relatively low amount of sequence data employed for tree building, the authors should also consider including additional sequence data where available and constraining the positions of certain lineages based on the results of other studies with larger datasets. For example, constraining Loponetta as sister to Anas (see e.g. Bulgarella et al., 2010), and placing Oxyura as the outgroup (see e.g. Gonzalez et al., 2009), though we recommend using more recent/robust phylogenetic hypotheses if available.

We, again, thank the reviewers for their excellent suggestion to constrain our tree to the topology published in Gonzalez et al., 2009 (see Figure 2—figure supplement 1, which shows the topology). After implementing this change we have resolved the polytomies in question that were previously present in the discarded NJ tree. We are unable to use other available sequence data in generating our tree, because there are no other common genes that are available for all species and all populations. The only available gene sequence for all low- and high-altitude taxa is the mitochondrial control region. The data sets in the citations below that were suggested by the reviewers unfortunately possess either a single low-altitude taxon and are missing the corresponding high-altitude taxon or vice versa. Nevertheless, we also ran the PIC analysis with branch lengths set to 1 to help account for potential variation in branch lengths across the tree, and this yielded qualitatively the same pattern of variation as our analyses using the actual branch lengths. Nevertheless, as described in our earlier response, we believe the paired-lineage Wilcoxon tests (rather than the PIC analysis) are the most appropriate for examining the convergent differences that exist across high-low pairs, and we have made changes throughout the manuscript to bring the results of the Wilcoxon tests to the forefront.

Bulgarella, M., Sorenson, M.D., Peters, J.L., Wilson, R.E. and McCracken, K.G., 2010. Phylogenetic relationships of Amazonetta, Speculanas, Lophonetta, and Tachyeres: four morphologically divergent duck genera endemic to South America. Journal of Avian Biology, 41(2), pp.186-199.

Unfortunately, this data set, while including many genes, does not include both high- and low-altitude populations and is missing 5 species from our data set within the genus Anas and the ruddy duck. We have, however, made many changes that improve our phylogenetic comparisons as stated above.

Gonzalez, Düttmann and Wink, 2009.

We have included a figure to outline the topology that we have used to constrain our newly created tree. This is the correct global waterfowl phylogeny to use as it contained all 7 species sampled in our study (see Figure 2—figure supplement 1).

4) The authors helpfully sampled from three sites in the muscle, at different depths, and took a mean of the activities from each location. Please report the degree of variation between the three locations.

We have included an additional table (Supplementary File 1J.) containing the enzymatic activities obtained at each depth of tissue. We have also made a small revision to the main text to describe the general pattern of variation observed between the three locations (subsection “Tissue Sampling”).

5) It would be helpful for all data to be presented in the manner of Figures 3, 4, and 5. Figure 6 is useful in showing differences in HK activity and ATPase activity in relation to evolutionary time, but the data in panel 6D, which only shows absolute rates of HK activity for high altitude groups, suggest that HK activity decreases below low altitude values as a function of time, whereas it is higher than in lowland counterparts (albeit not always significantly higher) for almost all species studied.

We have added graphs from HK and ATP synthase in Figures 3 and 4 to show the same data consistently across all enzymes. We have also made changes to the text to discuss HK and ATP synthase in the context of convergence (subsections “Convergence Across Multiple Pathways of Energy Metabolism” and “Some Metabolic Changes Arise Only After Prolonged Evolutionary Time at High Altitude”).

6) One of the main themes of the paper was that fatty acid oxidation capacity is increased at high altitude. However, this is based on a single enzyme indicator, HOAD, whereas other aspects of metabolism were assessed with multiple enzymes. This potential limitation should be noted or discussed.

This is a great point. HOAD is a good marker of beta-oxidation capacity, but the overall capacity for lipid oxidation will of course depend on other key regulators as well (e.g., mitochondrial fatty acid uptake by CPT). We’ve added two new sentences to address this issue (subsection “Convergence Across Multiple Pathways of Energy Metabolism”).

7) In the third paragraph of the subsection “Convergence Across Multiple Pathways of Energy Metabolism”, the change in complex II (succinate dehydrogenase) activity is attributed to a need to support greater capacities for fatty acid oxidation, though this is not correct. The free FADH2 from beta-oxidation does not donate electrons to complex II (which oxidizes succinate using a fixed FAD/FADH cofactor), but to the electron-transferring flavoprotein complex (CETF). Please clarify/revise.

We did not originally intend to suggest that FADH2 produced via beta-oxidation would be utilized by complex II (TCA cycle), but we agree in hindsight that this point was not clear. Apologies for the confusion, we have deleted this sentence from the paper.

8) Why are the results for Complex V (ATP synthase) not reported in the comparisons between populations/species, and included only in the analysis of time? They seem to be important.

We have added a short section to report changes in ATP synthase in the context of species-specific changes (subsection “Convergence Across Multiple Pathways of Energy Metabolism”).

9) The possible explanations for increased ATP-synthase activities at altitude are useful, but the second paragraph of the subsection “Some Metabolic Changes Arise Only After Prolonged Evolutionary Time at High Altitude” refers to anoxia, and does not appear to be relevant to non-anoxia tolerant waterfowl. The authors cite a review article here (Garcia-Aguilar and Cuevez) but the original work was carried out by St-Pierre and colleagues (2000, PNAS, 97(15), pp.8670-8674) in anoxia-tolerant frogs. The relevance to the current study is not clear, since it is not necessarily the case that membrane potential would fall in hypoxia exposed waterfowl (a different state to anoxia).

We agree with the reviewers that the relevance of this work is not clear for this study and removed this section from the Discussion.

10) The authors speculate that the differences in metabolic function are related to hypoxia or cold at altitude, but given this is a flight muscle, how might the decreased air resistance alter the metabolic demand placed on this muscle? Would activity be expected to fall as there is less drag, or might the low air pressure increase the metabolic demand of lift?

This is a great suggestion. In fact, recent work on high-altitude flight in the bar-headed goose done by Bishop et al., 2015, show that the reduction in drag has been shown to be more than offset by the reduction in capacity to produce lift at low air densities, and thus increases the metabolic cost of flight at high altitudes. We have included a sentence outlining this in the Introduction.

11) In the statistical analyses presented in Supplementary file 1B and C, there are highly significant species by altitude interactions, but these are not addressed. When there are significant interactions, main effects should not be considered until the interactions are addressed, either by plotting them out to see if they are all in the same direction and just have different slopes, or by separating the species and looking at them individually if there is crossover of the effects. The authors need to discuss what the interactions were and how they handled them. Most of these are present in cases where a significant altitude effect was detected, so they are potentially crucial to the conclusions.

The significant interaction effects were attributed primarily to opposing patterns observed in the ruddy ducks, with the exception of HK and ATPase, where the interaction were attributable to changes observed in either species that are more recently established (HK) or established for longer periods of time at altitude (LDH, ATPsyn). We have added mention of the significant interactions where we discuss these patterns of variation in the Results/Discussion section (subsections “Idiosyncratic Changes in High-Altitude Ruddy Ducks”, “Some Metabolic Changes Arise Only After Prolonged Evolutionary Time at High Altitude”, as well as “Enzyme activities and myoglobin assays”).